# Nutrient Status of Cucumber Plants Affects Powdery Mildew (*Podosphaera xanthii*)

**DOI:** 10.3390/plants10102216

**Published:** 2021-10-19

**Authors:** Yigal Elad, Dor Barnea, Dalia Rav-David, Uri Yermiyahu

**Affiliations:** 1Department Plant Pathology and Weed Research, Agricultural Research Organization, The Volcani Center, 68 Hamakabim Rd, Rishon LeZion 7534509, Israel; dor.barnea@gmail.com (D.B.); dalia@volcani.agri.gov.il (D.R.-D.); 2The Robert H. Smith Faculty of Agriculture, Food and Environment, The Hebrew University of Jerusalem, Rehovot 76100, Israel; 3Agricultural Research Organization, Gilat Research Center, D.N. Negev 2, Bet Dagan 85280, Israel; uri4@volcani.agri.gov.il

**Keywords:** agrotechnical control, calcium, *Cucumis sativus*, cultural control, integrated management, powdery mildew, magnesium, plant disease, potassium

## Abstract

We examined the effects of applications of N, P, K, Mg, and Ca through an irrigation solution and spraying K, Ca, and Mg salts on cucumber powdery mildew (CPM, *Podosphaera xanthii*) in potted plants and under commercial-like conditions. Spraying CaCl_2_ and MgCl_2_, or KCl and K_2_SO_4_, decreased CPM. There were significant negative correlations between the anion-related molar concentrations of the salts and disease severity. Among the sprayed treatments, NaCl provided significantly less CPM control when applied at a low (0.05 M) concentration, as compared with CaCl_2_ and MgCl_2_. When sprayed applications of Mg and K salts were analyzed separately from the untreated control, the Cl^−^ salts were found to be more effective than the SO_4_^−2^ salts. High N and Mg concentrations in the irrigation water delivered to young, fruit-less cucumber plants reduced CPM, whereas more CPM was observed when the irrigation solution contained a medium amount of P and a high amount of K. In contrast, mature, fruit-bearing plants had less severe CPM at higher N, lower P, and higher K levels. Spraying mature plants with monopotassium phosphate, polyhalite (K_2_Ca_2_Mg(SO_4_)_4_·2H_2_O), and the salts mentioned above over an entire growing season suppressed CPM. CPM severity was also reduced by spray applications of Ca, Mg, and KSO_4_^−2^ and Cl^−^ salts. Spray applications provided better CPM control than fertigation treatments. Induced resistance is probably involved in the effects of nutrients on CPM.

## 1. Introduction

*Podosphaera xanthii* (syn.: *Podosphaera fuliginea*, *Sphaerotheca fuliginea*, *Sphaerotheca fusca*) is one of the causal agents of powdery mildew of cucurbits [1]. It is a biotrophic pathogen that is distributed by asexual conidia that germinate on green plant organs (mainly leaves), producing germ tubes that form appressoria. The appressoria form penetration pegs that penetrate the epidermal cells and then form haustoria in close contact with the host cells. The pathogen grows over the leaf surface and superficially inside the epidermis and forms chains of conidia attached to conidiophores, which are visible as white marks on the leaves [2]. The white marks spread from older to younger leaves and the cucumber powdery mildew (CPM) severity is measured according to the percentage of leaf surface covered in white [3]. CPM is managed using fungicide sprays and, to a lesser extent, by biocontrol; partial resistance is also available in some cultivars [3,4,5].

Mineral nutrients are needed for plant development and play important roles in all plant-life processes. Nitrogen, phosphorus, and potassium are essential for biomass production and plant growth. Calcium and magnesium are essential for many cellular processes and are less important for plant growth [6]. Microelements are important for plant life but are beyond the scope of the present publication. Therefore, we will review only their roles in plant diseases. Minerals are important in plant–pathogen interactions [7,8,9]. All essential mineral nutrients affect the progress of plant diseases [8,9]. However, those effects vary by microelement and between different plant–pathogen systems [10]. Different nutrients’ concentrations influence the resistance of plants to pathogens through metabolic changes, changes in the surrounding conditions, effects on the activity of enzymes, the synthesis of cell walls, the cell membrane’s permeability, the synthesis of amino acids, polyphenols, and sugars [11], and plant resistance gene upregulation, as was demonstrated for foliar applications of N, K, Mn, and Zn [7].

The effects of N on plant disease depend on the form of N, the species and organ of the affected plant, and the pathogen’s identity [12]. It is generally understood that obligate plant parasites thrive in the presence of excess N fertilization. This has been attributed to the associated increased proliferation of young plant tissue and to the increase in the amino acid apoplast concentration on the leaf surface that encourages conidia germination and penetration [13]. A high N concentration in tomato plants increased *Fusarium oxysporum* f. sp. *lycopersici* severity and decreased the expression of phenylpropanoid pathway enzymes and the levels of lignin, callose, tylose, and rishitin, which all play important roles in plant defense [12]. Grape downy mildew increased and the synthesis of the phytoalexin resveratrol decreased at higher concentrations of N [14]. Rust and downy mildew of cereals increased when high levels of NO_3_^−^ were applied as fertilizer [15]. In sweet basil, the necrotrophs *Botrytis cinerea* and *Sclerotinia sclerotiorum* are more prevalent when there are higher concentrations of N in the plant shoots [16,17]. Increasing the total N concentration in the irrigation solution was shown to increase the level of sweet basil downy mildew in potted sweet basil plants as well as sweet basil grown under field conditions. Moreover, when NH_4_^+^ accounted for a large proportion of the total N in the irrigation solution, less severe downy mildew was observed [18].

Phosphorus has been reported to decrease soilborne diseases such as take-all in wheat (*Gaeumannomyces graminis* in *Triticum aestivum*), charcoal rot (*Macrophomina phaseolina*) in maize (*Zea mays*) [9] and soybean (*Glycine max*) [19], and *Verticillium dahlia* wilt in potato (*Solanum tuberosum*) [20], as well as the foliar disease powdery mildew in wheat [21]. P fertigation has been found to increase the severity of onion downy mildew (*Peronospora destructor*) [22]. It has also been argued that P plays a role in induced resistance in some pathosystems [23,24,25].

Potassium at optimal tissue concentrations decreases the susceptibility of plants to disease [9]. K reduced the incidence of disease caused by the ascomycete pathogens *B. cinerea* and *S. sclerotiorum* [16,17]. K also suppressed downy mildew (*Peronospora plantaginis*) severity in *Plantago ovata*, the plant used to produce isabgol [26], and suppresses sweet basil downy mildew when it is sprayed on the plant [27].

Calcium plays a role in plant defense systems [28]. It is an important constituent of pectin and other cell-wall components. Ca binds pectin oligomers and by that it prevents pathogen penetration. Ca acts against cell-wall-degrading enzymes of pathogens [28,29]. Ca also plays a role in the activity of pathogenesis-related proteins, the expression of defense-related genes, and hypersensitive reactions [30,31,32]. Ca increases the resistance of plants to pathogens such as species of *Pythium*, *Sclerotinia*, *Botrytis*, and *Fusarium* [9]. Ca reduced in sweet basil the incidence of *B. cinerea* and *S. sclerotiorum* [16,17]. Sprayed applications of Ca reduced the severity of downy mildew (*Sclerospora graminicola*) in pearl millet (*Pennisetum glaucum*) [33]. Irrigation with increased levels of CaCl_2_ decreased sweet basil downy mildew under field conditions [27].

Magnesium affects plant diseases, directly and indirectly, through its antagonistic interactions with other nutritional minerals (e.g., K, Ca, and Mn) [34]. *Fusarium oxysporum* f. sp. *conglutinans* disease in cotton (*Gossypium arboreum*) was reduced when Mg availability was at optimal levels. A high Mg concentration that interfered with Ca absorption increased the severity of bacterial speck (*Xanthomonas campestris* pv. *vesicatoria*) in tomato (*Solanum lycopersicum*) [34]. Mg decreased downy mildew (*Peronospora arborescens*) severity in poppy (*Papaver somniferum*) [35] and decreased tobacco downy mildew (*Peronospora tabacina*) severity [36]. Irrigation with increased MgCl_2_ also decreased sweet basil downy mildew under field conditions [27].

There have been previous studies on nutritional elements’ effects on diseases of cucumber. Fertigation of cucumber with P in a hydroponic system and spray applications of P-containing salts were found to decrease powdery mildew [24,37,38]. Spray applications of Ca and K reduced gray mold (*B. cinerea*) on fruits and stems of cucumber plants and K fertigation reduced downy mildew (*Pseudoperonospora cubensis*) on cucumber leaves [39]. Nitrate was found to protect cucumber plants against *Fusarium oxysporum* f. sp. *cucumerinum* [40]. Monopotassium phosphate (MKP) is effective against cucumber powdery mildew under commercial conditions [41,42].

We examined the effects on CPM of salts containing the cations Ca, Mg, and K and the anions Cl and SO_4_ applied as foliar sprays (alone or in combination) to young plants and mature, fruit-bearing plants. We also examined the application of N, P, and K as well as Ca and Mg, which were applied via irrigation water. In this study, we first tested in potted plants the effects of spray treatments and determined the optimal concentrations for the application of each of the mineral cations. Later, we tested the effects of a limited number of concentrations and combinations of nutrients on CPM severity under commercial-like conditions.

## 2. Results

### 2.1. Effect of Sprayed Applications of Salt Solutions on CPM in Potted Cucumber Plants (Expt. A-s1)

Sprayed applications of Ca and Mg Cl^−^ salts and K salts with Cl^−^ and SO_4_^−2^ at concentrations of 0.5 to 1.0% (Table 1) were made to mature cucumber plants. CPM developed on the leaves and was significantly suppressed by most of the salts (Figure 1a). The concentrations of CaCl_2_ and KCl did not affect their suppressive effects. We evaluated the relationships between CPM severity and the molar concentrations of the applied salts according to the cations and the anions of the salts. A significant negative correlation was observed between the anion-related molar concentrations of the salts and disease severity (Figure 1b), whereas the relationship between the cation molar weight of the salts and CPM severity was insignificant (data not presented). Thus, in our subsequent experiments, we considered the anion-related molar concentrations of the salts.

### 2.2. Effect of Sprayed Applications of Cl^−^ Salt Solutions on CPM in Potted Plants (Expts. B-s1 and B-s2)

To evaluate the effects of different concentrations of spray-applied Cl^−^ on CPM, chloride salts of Ca, Mg, and Na were applied to cucumber plants (Table 1, Figure 2). In one experiment, only NaCl was applied (Expt. B-s1, Figure 2a,b). In a second experiment (Expt. B-s2, Figure 2c), Cl salts of Na, Ca, and Mg and a mixture of Ca and Mg salts were applied. All of the salts significantly reduced CPM severity (Figure 2a–c). The relationship between the concentration of Cl^−^ in the spray solution and the Cl^−^ concentration in the leaves was positive and significant (Figure 2a) and the Cl^−^ concentration in the leaves was strongly negatively correlated with CPM severity (Figure 2b). NaCl was significantly less effective at the lower 0.05 M concentration (Figure 2c). Interestingly, the relationship between the degree of suppression and the salt concentration was similar across the different salts, with the greatest control observed at salt concentrations of less than or equal to 0.1 M and no further disease reduction observed at the higher concentrations. Furthermore, the combination MgCl_2_+CaCl_2_ did not result in an additive effect, as compared with each salt alone (Figure 2c).

### 2.3. Effects of Spray Applications of Cl^−^and SO_4_^−2^ Salt-Solutions and a Fungicide on CPM in Potted Plants (Expts. B-s3 and B-s4)

In Expt. B-s3, Mg and K salts containing either Cl^−^ or SO_4_^−2^ anions were applied at two concentrations (Table 1, Figure 3a). The salts effectively suppressed CPM; disease severity in the control treatment reached 86%, while all of the salt treatments suppressed the disease severity down to 0.7 to 19.4%. When the salt-spray treatments were analyzed separately from the untreated control, the Cl^−^-containing salts were statistically more effective than the SO_4_^−2^-containing salts and the higher salt concentrations were more effective than the lower salt concentrations (*p <* 0.0009, statistical analysis not presented). In these experiments, there was no difference between K- and Mg-containing salts. In other experiments (Expt. A-s4), all of the tested salts except for KCl provided CPM suppression that was just as good as that provided by the chemical fungicide penconazole (Figure 3b). 

### 2.4. Pot Experiments—Supplemental Nutrients in the Fertigation Solution under Controlled Conditions (Expts. B-#-f)

Effects of N, P, and K applied through the fertigation solution on CPM severity (Expts. B-N/P/K-f).

Different concentrations of N, P, and K in the irrigation solution were achieved without changing the concentrations of the other major ions, as described in the Materials and Methods section and Table 1.

*N (Expts B-N-f)*: Raising the concentration of N in the fertigation solution (15% NH_4_^+^) from 0.7 to 14.3 mM (Table 1) resulted in a gradual increase in the N concentration in the cucumber leaves, up to 3.91% of the leaf dry weight (Figure 4a). Increasing the N concentration in the irrigation solution resulted in a minor decrease in CPM severity (Figure 4b). Similarly, the major increase in N leaf concentration was associated with a minor decrease in disease severity (Figure 4c).

*P (Expts. B-P-f)*: Raising the concentration of P in the fertigation solution from 0 to 0.65 mM (Table 1) gradually increased the P concentration in the cucumber leaves, up to 0.8% of the leaf dry weight (Figure 5a). Increasing the P concentration in the irrigation solution led to a peak in CPM severity at 0.32 mM P (Figure 5b). Similarly, the change in the leaf concentration of P led to a peak in disease severity at 0.46 mM P (Figure 5c). *K (Expts. B-K-f)*: Raising the concentration of K in the fertigation solution from 0.3 to 2.6 mM (Table 1) gradually increased the K concentration in the cucumber leaves, up to 3.16% of the leaf dry weight (Figure 6a). Increasing the K concentration in the irrigation solution led to a gradual increase in CPM severity (Figure 6b). Similarly, the change in K leaf concentration led to a gradual increase in disease severity (Figure 6c).

### 2.5. Relationship between CPM Severity and Ca and Mg Supplied through the Fertigation Solution (Expts. B-Ca/Mg-f)

*Ca (Expts B-Ca-f)*: Raising the concentration of Ca in the fertigation solution from 1.0 to 4.0 mM resulted in a gradual increase in the Ca concentration in the cucumber leaves, up to 4.29% of the leaf dry weight (Figure 7). Ca levels in the irrigation solution and leaves did not significantly affect CPM severity (data not shown).

Mg *(Expts B-Mg-f)*: Raising the concentration of Mg in the fertigation solution from 0.82 to 4.94 mM resulted in a gradual increase in the Mg concentration in the cucumber leaves, up to 1.94% of the leaf dry weight (Figure 8a). Increasing the Mg concentration in the irrigation solution led to a gradual decrease in CPM severity (Figure 8b). Similarly, the change in Mg leaf concentration resulted in a gradual decrease in disease severity (Figure 8c).

### 2.6. Effects of Spray Applications of Various Salts on CPM under Semi-Commercial Conditions (Experiments B-SCs-a/b)

The semi-commercial experiments involved mature, fruit-bearing plants and included: (a) sprays of MKP and polyhalite (Expt. B-SCs-a, Table 1), which are commercially available as complex fertilizers and contain K or Mg, Ca, and S, respectively; and (b) MgCl_2_, CaCl_2_, and K_2_SO_4_ at 0.05 and 0.1 M (Cl^−^ or SO_4_^−2^-related) and a combination of 0.05 M of these salts and 0.1 M K_2_(SO_4_) (Expt. B-SCs-b, Table 1). CPM severity was significantly reduced by MKP and polyhalite at the three evaluation times (Figure 9a) and in terms of the AUDPC (Figure 9b). The salts generally reduced CPM severity at the three evaluation times and the calculated AUDPC. The calculated AUDPC revealed better disease suppression by the combination of the salts as compared with applications of the same concentrations of the individual salts (Figure 10a,b).

### 2.7. Commercial-Like (CL) Net-House Experiments to Test the Effects of Fertigation and Spray Applications

N, P, and K fertigation was tested under commercial-like conditions (Expt. CL1) with mature plants over a whole growing season (Table 1). A somewhat lower CPM severity was observed among the plants that were fertigated with low N (2.9 vs. 7.1 and 14.3 M N, Figure 11a1) and high P (0.65 vs. 1.29 M P, Figure 11a2). A lower CPM severity was observed among plants treated with the low concentration of K (1.0 vs. 2.6 and 5.1 M K, Figure 11a3). In the same experiment, the fungicide and the mixture of MgCl_2_ and K_2_SO_4_ provided significant CPM control (57.9 and 75.6%, respectively; Figure 11).

Except for the low-N treatment, the respective cumulative number of fruits and the weight of the yield per plot in the various fertigation treatments were similar and ranged between 10.8 and 13.2 fruits/plant and between 1.226 and 1.356 kg/plant, respectively. In contrast, the low-N treatment yielded significantly less than the rest of the fertigation treatments, with a cumulative 8.8 fruits/plot and 0.908 kg/plant. The three spray treatments in Expt. CL1 had cumulative yields of 11.0–11.8 fruits/plant and 1.115–1.235 kg/plant, with no significant differences between them.

Spray treatments were also applied in the second commercial-like experiment (Expt. CL2). CPM severity was significantly suppressed by MKP and the fungicide triadimenol: 75.6 and 61.8%, respectively. MgCl_2_ provided better CPM control (97.6% disease reduction). The control provided by the mixed-salts treatment (73.2% disease reduction) was no better than that provided by each salt alone (Figure 12). The various treatments yielded 12.1–14.3 fruits/plant and 1.2–1.6 kg fruits/plant, with no significant differences between the treatments.

## 3. Discussion

A great deal of information is available in the literature on the effects of mineral nutrition on the development of plant diseases (e.g., [43]). It is widely recognized that nutrition can influence disease, but much of the available information is contradictory, with both disease suppression and disease promotion described [43,44]. Effects of individual ions and combinations of ions on plant diseases have been demonstrated. In soybean (*Glycine max*), K, Ca, Mg, S, and Fe all induce resistance to *Fusarium oxysporum* infection [45]. As mentioned above, we observed an effect on disease by Ca, Mg, and K treatments [18], by Mn and Zn treatments [46], and by N in the irrigation solution [27]. In the present study, we tested the effects of N, P, K, Ca, and Mg ions applied as part of the irrigation solution and Ca, Mg, and K salts applied in spray solutions.

### 3.1. N Involvement

There is a general notion that high levels of N will increase the severity of diseases that are caused by obligate parasites [7]. In this study, increasing the N concentration in the irrigation water somewhat decreased the severity of CPM in younger plants that had no fruit and increased the disease severity in mature, fruit-bearing plants that were grown under commercial-like conditions. Similar to our results for young cucumber, a previous study found that N (300 mg/L) applied with an irrigation solution to potted plants under greenhouse conditions reduced the severity of downy mildew of cucumber (*Pseudoperonospora cubensis*) by 24% [47]. Similar to our results with mature plants, higher N concentrations in the irrigation water were previously shown to lead to more severe downy mildew in sweet basil under commercial-like conditions [27]. Wang et al. [40] reported that nitrate significantly suppressed Fusarium wilt caused by *F. oxysporum* f. sp. *cucumerinum* (FOC), as compared with ammonium, in both pot and hydroponic experiments. The researchers found that nitrate protects cucumber plants against *F. oxysporum* by decreasing root citrate exudation and FOC infection, suggesting a role of N in disease suppression in cucumber. 

Previous studies have found that P affects disease levels differently in different pathosystems. Mustafa et al. [21] found that P inhibits wheat powdery mildew (*Blumeria graminis*), whereas Develash and Sugha [22] found that P increases the severity of onion downy mildew (*Peronospora destructor*). Although we have researched the effects of plant nutrients in various pathosystems, the present study is the first to study the effect of P on a crop disease. We found that P increased the severity of CPM among both young plants and mature, fruit-bearing plants.

### 3.2. K Involvement

In the present study, K was applied as part of the irrigation solution and also as salts in a spray solution. Increasing the K concentration in the irrigation water increased CPM in young plants, but decreased disease severity in older plants and when it was applied as sprayed salts. The different effects of K observed in the present study suggest that K might also have different effects in the pathosystems that have been examined, but further investigation into the age effect on the response to K is needed. Sweet basil downy mildew, which is caused by an obligate parasite (*P. belbahrii*), was reduced by spray-applied K, but not by K that was applied as part of the irrigation solution [18]. In earlier studies, the necrotrophs *Botrytis cinerea* and *Sclerotinia sclerotiorum* were suppressed by K applied as part of the irrigation water or as part of a spray solution [16,17].

The application of K_2_SO_4_ reduced the downy mildew (*Plasmopara viticola*) incidence on grapevine (*Vitis vinifera*) leaves. An increased K concentration in grapevine petioles increases the constitutive and post-inflectional accumulation of phenolic acids, such as o-coumaric acid and p-coumaric acid, and of total phenols [48]. K was also associated with increased leaf activity of the phenylalanine ammonia-lyase and increased disease resistance [48]. The addition of K to a low-nitrate fertilizer reduced cucumber fruit gray mold (*B. cinerea*) and stem infections [49]. In addition, foliar-applied KNO_3_ reduced the incidence and severity of cotton Alternaria leaf blight (*Alternaria macrospora* and *A. alternata*) [49].

Application of 4–30 mM KNO_3_ before inoculation greatly reduced the incidence of stem rot (*Phytophthora sojae*) in soybean. The extent of that disease reduction was related to the increased K levels in the plants, particularly in plants’ cortex layer [50]. Dipotassium hydrogen phosphate reduced downy mildew (*S. graminicola*) of pearl millet (*Cenchrus americanus* syn. *Pennisetum glaucum*) under experimental and commercial-greenhouse conditions. However, unlike our findings, the disease suppression in that system was related to the phosphate component [51]. Contrasting results have been reported for KCl fertilization, which reduced the severity of wheat leaf rust (*Puccinia triticina*); however, that response may have been partially related to the chloride in the KCl fertilizer [23].

Potassium enrichment in cucumber reduced the natural incidence of downy mildew (*Pseudoperonospora cubensis*) [39]. A reduction in cucumber powdery mildew was achieved through the application of potassium dihydrogen phosphate, magnesium sulfate, ferrous sulfate, and potassium monohydrogen phosphate [41]. Effective control of cucumber powdery mildew, as expressed by a 99% reduction in symptoms, was achieved one or two days post application of a single spray of phosphate and K salt solutions and those treatments were also effective at a later stage [37,38]. The authors of those works concluded that these properties of potassium and phosphate salts make them appropriate for use as foliar fertilizers with potential beneficial effects on disease control.

### 3.3. P Involvement

MKP has been reported to be beneficial for the control of powdery mildews. MKP sprayed alone, mixed with, or in alternation with fungicides suppressed powdery mildews of apple, peach, and nectarine (*Podosphaera leucotricha* in *Prunus persica* var. *nucipersica*), grapevine (*Uncinula necator*), rose (*Sphaerotheca pannosa* in *Rosa × hybrida*)*,* cucumber and melon (the fungus named in that research was *S. fuliginea*; the plants were *Cucumis sativus* and *C. melo*, respectively), and mango (*Oidium mangifera* in *Mangifera indica*) [42]. In addition, nutrient solutions containing P at concentrations of 5 to 40 ppm in a hydroponic system induced systemic resistance against *S. fuliginea* in young cucumber plants. Once the pathogen had been established, root-applied P did not affect well-developed colonies. However, foliar application of a 1% solution of MKP effectively protected the foliage against powdery mildew, regardless of the P concentration in the water [24]. In the present work, we describe the suppression of CPM by MKP under semi-commercial and commercial-like conditions.

### 3.4. Ca and Mg Involvement

Adding Ca to the irrigation solution reduced gray mold on fruits and stems of cucumber plants. Ca also reduces gray mold in pepper (*Capsicum annum*) and eggplant (*Solanum melongena*) [49]. Seed treatment with 90 mM CaCl_2_ suppressed pearl millet downy mildew (*Sclerospora graminicola*) and reduced the biomass of the pathogen in the treated plants [33]. Chardonnet and Donèche [52] suggested that Ca treatment of cucumber fruit prior to infection can increase the cell-wall-bound Ca and thereby decrease pectin digestion by fungal pectinolytic enzymes and fruit infection by the necrotrophic fungus *Botrytis cinerea*. Protection against cucumber powdery mildew in a greenhouse was obtained with the application of calcium nitrate and potassium phosphate in doses of 20 g/L and a 6 mL/L spray [53]. In the present work, we were able to increase the Ca concentration in the cucumber plants, but this did not affect the level of CPM.

Tomato (*Solanum lycopersicum*) root and crown rot (F. *oxysporum* f. sp. *radicis-lycopersici*) was reduced by Ca(NO_3_)_2_ but not by MgSO_4_; the disease reduction was related to the level of nitrate [54]. Mg and Ca suppressed sweet basil downy mildew in sweet basil [18]. In the present study, both Mg in the irrigation solution and Mg applied as a spray suppressed CPM.

Since the combination of salts gave only a slight improvement in CPM control over the control that was observed with the single treatments, we cannot at all generalize that a combination of ions or salts can provide any major improvement in disease control. In a recent study involving sweet basil downy mildew, Ca and Mg did not have an additive suppression effect on sweet basil downy mildew [18]. It has been suggested that the deleterious effect of Ca on Mg load in the canopy was the reason that the combination of supplemental Mg and supplemental Ca did not improve disease control when both elements were applied together. There is no evidence in the literature for any such effect of this cation combination on plant disease. Furthermore, the observed suppression of the downy mildew by Ca and Mg treatments suggests a general mode of action that is triggered by either of those cations (i.e., induced resistance).

Nevertheless, the mixture of ions in a spray solution was also tested using polyhalite (K_2_Ca_2_Mg(SO_4_)_4_^.^2(H_2_O)). Polyhalite effectively suppressed CPM severity. This seems to be the first report of any disease control provided by this fertilizer.

### 3.5. Role of Anions

Interestingly, the suppressive effects of spray-applied K, Ca, and Mg salts on CPM of young cucumber plants were correlated with the anion concentration and not with the cation concentration. The choice of anion to be used with the cations that were used was a question that was raised during the present research. Cl^−^ was more effective than SO_4_^−2^ in two sets of experiments (Expts. A-s1 and A-s3), and repeated experiments revealed a relationship between the Cl^−^ concentration and the efficacy of disease control (i.e., at concentrations greater than 0.05 and 0.1 M, no improvement in CPM control was observed). Nevertheless, at the 0.05 M concentration, NaCl was significantly less effective than a 0.05 M chloride concentration of Mg and Ca salts. Under commercial-like conditions, fertigation-applied MgCl_2_ suppressed sweet basil downy mildew better than fertigation-applied MgSO_4_. Cl^−^ was also the anion of choice in other greenhouse experiments [18].

There have been a few other reports on the effects of these anions on plant diseases. The effects of Cl fertilizer (e.g., KCl) on winter wheat were investigated in field trials, which revealed that the applied Cl suppressed the foliar diseases powdery mildew (*Erysiphe graminis*) and leaf rust (*Puccinia recondite*) [55]. Not much information is available about the effects of the anion on plant diseases. The application of KCl at 150 to 400 µg of K per gram of dry soil increased the incidence of Phytophthora root and stem rot (*Phytophthora sojae*) on susceptible soybean plants. That increased disease appears to be due to the presence of chloride [56].

## 4. Materials and Methods

### 4.1. Plants and Experiments

Cucumber (*Cucumis sativus*) cv. Bet Alpha seeds were planted in seedling trays containing 4 × 4 × 10 cm invert-pyramidal cells filled with perlite. The perlite (medium size, 1.2 mm) was washed in water prior to use (Agrifusia, Fertilizers & Chemicals Ltd., Haifa, Israel) for potted-plant experiments. The developed seedlings were transplanted to pots at 3 weeks of age. Seeds were also germinated directly in pots, as described below. To prevent damping-off in the pots harboring germinating seeds, we treated the growth medium once with a fungicide (0.25% Dynon in water drench, containing 722 g/L Propamocarb HCl, Bayer, Germany). Cucumber Cv. 501 was used in the commercial-like (CL) experiments, as described below.

The experiments involving plants grown in pots were performed at two sites in Israel: the Volcani Institute in Rishon LeZion (31º58′09″ N 34º48′02″ E) (Site A) and the Gilat Research Center in the northern Negev region (31º23′12″ N 34º43′15″ E) (Site B), 68 km south of Site A. Experiments were also carried out using plants grown in containers under commercial greenhouse conditions in a net-house at the Gilat Research Center (Site C). At Site A, experiments were carried out at an experimental greenhouse, in 2-L pots and 10-L pots (potted and semi-commercial conditions, respectively). At Site B, experiments were carried out in 2-L pots. The commercial-like experiments (Site C) were carried out in 136-L containers. For the fertigation experiments, cucumber plants were planted in pots or containers filled with perlite.

A potting mixture consisting of coconut fiber:tuff (unsorted to 8 mm; 7:3 vol.:vol.) was used in the experiments that involved foliar salt treatments. The plants were irrigated to excess via a drip system two to four times a day, depending on the season, at a volume calibrated to lead to >30% water leaching. The daily irrigation volume was determined after analyzing the irrigation and drainage solutions once every 2 weeks to prevent over-salinization or acidification of the root-zone solution. Plants in pots and containers were maintained according to the local extension service’s recommendations. All pot experiments were irrigated with fresh water (electrical conductivity (EC): 0.4 dS/m = desalinated water). The tested elements were applied with the irrigation solution (fertigation) or as a foliar spray with no added surfactants, as described below and as summarized in Table 1.

### 4.2. Pathogen and Disease

Cucurbit powdery mildew (*Podosphaera xanthii*, CPM) occurred naturally and the cucumber cultivar used is known to be susceptible to CPM. Nevertheless, to ensure even distribution of the pathogen across the plants, every potted-plant experiment was artificially inoculated with a conidial suspension (10^4^ conidia/mL) using a hand sprayer that produced a fine mist that dried within 10 min. The pathogen formed the typical white symptoms of CPM on leaves and disease severity was evaluated according to the symptoms coverage area of the leaves, on a 0 to 100 scale, in which 0 = healthy leaves and 100 = leaves completely covered by disease signs/symptoms. The evaluated leaves were as follows: Leaf 5 from the plant base as a mature leaf; Leaves 6–9 as medium leaves; and Leaves 9–12 as younger leaves. For whole-plant CPM severity, the average ratings for the evaluated leaves on each plant were calculated. The area under the disease progress curve (AUDPC) was calculated throughout the period of epidemic development.

### 4.3. Foliar Application of Salt Solutions to Potted Cucumber Plants (Experiments A-s 1 and B-s 1–4)

Cucumber seedlings were transplanted into pots containing perlite or growth mixture, as described above, unless noted otherwise. Fertigation was carried out with the fertilizer 4-2-6 (N-P_2_O_5_-K_2_O) + 3% microelements (Fertilizers and Chemical Compounds Ltd., Haifa, Israel) throughout the experiments. Spray with water with no salt served as an untreated control. In young cucumber plants, following the formation of two or three leaves, when the plants had reached a height of 20 to 30 cm (Expt. B-s) and in mature plants bearing at least 18 leaves (Expt. A-s), the foliar treatments were initiated and plants were artificially inoculated with conidia of *Podosphaera xanthii*. Experiments A-s 1 and Experiments B-s 1 through 4 were each repeated twice, with five replicates each time, and arranged in randomized blocks.

Sprays were conducted twice a week with a hand sprayer that formed a mist of small drops and contained up to 1 L of water. Solutions contained 0.1 and 1.0% of K_2_SO_4_, KCl, MgCl_2_, and CaCl_2_ with various molar concentrations of the applied ions (Expt. A-s1): NaCl at concentrations of 0, 0.05, 0.1, 0.2 0.4, and 0.6 M (Cl^−^) (Expt. B-s1); MgCl_2_, CaCl_2_, NaCl, and MgCl_2_ with CaCl_2_ at concentrations of 0, 0.05, 0.1, 0.2, and 0.4 M (Expt. B-s2); MgSO_4_, MgCl_2_, K_2_SO_4_, and KCl at anion concentrations of 0.1 and 0.4 M (Expt. B-s3); MgSO_4_, MgCl_2_, K_2_SO_4_, and KCl at anion concentrations of 0.2 M; and the fungicide penconazole (200 g/L as Ofir2000, Syngenta, Switzerland) sprayed with a 0.035% formulation once a week (Expt. B-s4).

### 4.4. Effects of Different Concentrations of N, P, K, Ca, and Mg in the Fertigation Solution on CPM (Experiments B-#-f)

Pot experiments were conducted in an unheated, polyethylene-covered greenhouse located at Site B. The aim of these experiments was to study the effects of different N, P, K, Ca, and Mg concentrations in the fertigation solution (“f” treatments) on the development of CPM in potted cucumber plants. The cucumber plants were planted in 2-L, perlite-filled pots with one plant per pot, in 10 replicates. Each set of cation concentrations was repeated twice, and pots were arranged randomly. The plants did not bear any fruit.

Nutrient solutions were prepared in 500-L containers containing all of the added nutrients. All of the plants were fertigated with 5-3-8 (N-P_2_O_5_-K_2_O) fertilizer (Fertilizers and Chemical Compounds Ltd., Haifa, Israel) for 2 weeks, until plant establishment. Later on, the effect of cation concentration was tested by tailoring the fertigation solution to each N, P, K, Ca, and Mg concentration of interest, as described below. The concentrations of nutrients that were not part of the experiments and remained the same across all treatments were as follows: 5.7 mM N (90% NO_3_^−^-N and 10% NH_4_^+^-N, excluding Experiment B1-N below), 0.352 mM P (excluding Experiment B1-P below), 2.6 mM K (excluding Experiment B1-K below), 1.3 mM Ca (excluding Experiment B1-Ca below), 0.54 mM Mg (excluding Experiment B1-Mg below), 1.1 mM SO_4_^−2^, 0.023 mM B, 9.8 µM Fe, 4.9 µM Mn, 2.1 µM Zn, 0.31 µM Cu, and 0.16 µM Mo. Solutions were prepared by dissolving KH_2_PO_4_, K_2_SO_4_, KNO_3_, NH_4_H_2_PO_4_, NaNO_3_, and NH_4_NO_3_ in water [17,18,27]. In Experiment B1-Ca-f, Ca was applied as CaCl_2_ and, in Experiment B1-Mg-f, Mg was applied as MgCl_2_. The EC in the different B-N/P/K-f treatments was 1.03–1.36 dS/m and the pH of the fertigation solution was 6.99–7.42. The EC in the different B-Ca/Mg-f treatments was 0.99–2.50 dS/m and the pH of the fertigation solution was 6.43–7.12.

#### 4.4.1. N Concentration in the Fertigation Solution (Experiments B-N-f)

The aim of these experiments was to study the effect of the N concentration in the fertigation solution on the development of CPM in potted cucumber plants. To characterize the response of cucumber plants to different concentrations of N in the fertigation solutions, six N concentrations (0.7, 1.4, 2.9, 5.0, 7.1, and 14.3 mM) were used while the concentrations of the other nutritional elements were kept constant.

#### 4.4.2. P Concentration in the Fertigation Solution (Experiments B-P-f)

The aim of these experiments was to study the effect of the P concentration in the fertigation solution on the development of CPM in potted cucumber plants. To characterize the response of cucumber plants to different concentrations of P in the fertigation solutions, four P concentrations (0, 0.065, 0.323, and 0.645 mM) were used while the concentrations of the other nutritional elements were kept constant.

#### 4.4.3. K Concentration in the Fertigation Solution (Experiments B-K-f)

The aim of these experiments was to study the effect of the K concentration in the fertigation solution on the development of CPM in potted cucumber plants. To characterize the response of cucumber plants to different concentrations of K in the fertigation solutions, five K concentrations (0.3, 0.5, 1.0, 1.8, and 2.6 mM) were used while the concentrations of the other nutritional elements were kept constant.

#### 4.4.4. Ca Concentrations in the Fertigation Solution (Experiment B-Ca-f, CaCl_2_ Supplement)

The aim of these experiments was to study the effect of Ca concentration on the development of CPM in potted cucumber plants. To characterize the response of cucumber plants to different concentrations of Ca (applied as Cl^−^ salt) in the fertigation solutions, without changing the concentrations of the other major ions (Table 1), four Ca concentrations (1.0, 2.0, 3.0, and 4.0 mM) were used while the concentrations of the other nutritional were kept constant.

#### 4.4.5. Mg Concentrations in the Fertigation Solution (Experiment B-Mg-f, MgCl_2_ Supplement)

The aim of these experiments was to study the effect of the Mg concentration in the fertigation solution on the development of CPM in potted cucumber plants. To characterize the response of cucumber plants to different concentrations of Mg (applied as Cl^−^ salt) in the fertigation solutions, without changing the concentrations of the other major ions (Table 1), four Mg concentrations (0.82, 1.65, 3.29, and 4.94 mM) were used while the concentrations of the other nutritional elements were kept the same for all treatments.

### 4.5. Foliar Application of Salt Solutions to Cucumber Plants Grown under Semi-Commercial Conditions (Experiments A-SC a and b)

At Site A, cucumber seedlings were transplanted into 10-L pots containing growth mixture. Fertigation was carried out with the fertilizer 4-2-6 (N-P_2_O_5_-K_2_O) + 3% microelements (Fertilizers and Chemical Compounds Ltd., Haifa, Israel) throughout the experiments. A spray with water with no salt served as an untreated control. Mature plants bearing at least 20 leaves, as well as flowers and fruits, were used. The foliar treatments were initiated when plants were artificially inoculated with conidia of *Podosphaera xanthii*. Experiments were conducted twice with six replicates each time and arranged in randomized blocks.

Spray treatments were applied twice a week with a hand sprayer containing up to 1 L of water and forming a mist of small droplets. The experiments included the treatments of MKP (Fertilizers and Chemical Compounds Ltd., Haifa, Israel) at 1% as recommended and polyhalite (K_2_Ca_2_Mg(SO_4_)_4_^.^2(H_2_O) (polysulfate, Fertilizers and Chemical Compounds Ltd., Haifa, Israel) [57] at concentrations of 0.125 and 0.25% (Expt. A-SC a). The salts MgCl_2_, CaCl_2_, and K_2_SO_4_ were sprayed at concentrations of 0.05 and 0.1 M (anion) and a combination of MgCl_2_ 0.05 M (Cl), CaCl_2_ 0.05 M, and K_2_SO_4_ 0.1 M (SO_4_^−2^) resulting in a total of 0.1 M Cl^−^ and SO_4_^−2^ (Expt. A-CS b).

### 4.6. Commercial-Like (CL) Net-House Experiments to Test the Effects of Minerals Applied as Part of the Fertigation Solution and/or as Spray Treatments

At Site C, experiments were carried out in a net-covered greenhouse (net = 50 mesh) having natural conditions with fluctuating RH and temperature. One experiment was performed to test the effects of fertigation treatments coupled with spray treatments (Expt. CL1). The second experiment was conducted to evaluate the effects of different spray treatments (Expt. CL2). Cucumber plants (Line 501, Hazera Genesis, Berorim, Israel) were maintained in a commercial nursery (Shetil Neto, Gevaram, Israel), transplanted on 1 March 2019 to the perlite boxes mentioned below, and grown until the end of May 2019, at which point they were all bearing fruit.

Cucumber plants were planted in perlite (medium size, 1.2 mm, Agrifusia) growth medium in polystyrene containers (1.0 × 0.8 × 0.17 m), with six plants per container and 12 plants in an experimental plot. Plants were irrigated daily according to local extension service recommendations. During the initial 5 days, plants were sprinkler-fertigated with 4.3 mM N (10% NH_4_^+^), 1.6 mM K, and 0.65 mM P in the fertigation solution to aid their establishment. After that initial period, the plants were irrigated through drippers and fertilized with 8.57 mM N, 0.65 mM P, and 3.2 mM K in water for 2 weeks until the fertigation treatments were initiated as described below (Table 2). Fertigation was performed from 1000-L tanks dedicated to each treatment, with a 17-mm drip-irrigation pipe that had a 2 L/h dripper embedded every 20 cm along its length. Plants were irrigated three times a day. There were six fertigation treatments with three N concentrations (2.9, 7.1, and 14.3 M), two P concentrations (0.65 and 1.29 M), and three K concentrations (1.0, 2.6, and 5.1 M). Each time there was any change in the concentration of one nutritional element, the other two elements were kept at the same ‘medium’ concentrations, so that only the concentration of one element was changed in each fertigation treatment (Table 2).

Each of the fertigation treatments was combined with one of the spray treatments: water, the fungicide triadimenol (250 g/L as Bayfidan, Bayer, Germany) sprayed at a concentration of 0.05% once every 2 weeks, or the combined salt treatment of MgCl_2_+K_2_SO_4_ 0.1 M (Cl^−^) + 0.1 M (SO_4_^−^). Experiment CL2 involved spray applications of MgCl_2_ 0.1 M (Cl^−^), K_2_SO_4_ 0.1 M (SO_4_^−^), MgCl_2_+K_2_SO_4_, 1% MKP, and the fungicide triadimenol, as mentioned above.

In both experiments, spray treatments were applied with a backpack sprayer equipped with a conical nozzle. Sprays were administrated until runoff every 3 to 5 days. The yield was harvested once a week starting from 40 days after planting and until the end of the experiments. The cumulative number of fruits and the weight of the yield were recorded. Experiment CL1 was organized in randomized split plots with four replicates for each combination treatment. Experiment CL2 was organized in randomized blocks with four replicates. Each plot consisted of two perlite boxes, with six plants in each box arranged in two rows, so that each plot had 12 plants.

### 4.7. Element Analysis

For all experiments, fully expanded mature leaves were sampled randomly at harvest time and used to determine leaf mineral concentrations. The leaves were rinsed with distilled water and dried in an oven at 70 °C for 48 h. The dried plant material was ground and subjected to chemical analysis. N, P, and K concentrations in the leaves were analyzed after digestion with sulfuric acid and peroxide [58]. Ca, Na, and Mg levels were analyzed after digestion with nitric acid and perchlorate [59]. The concentrations of N and P were determined with an autoanalyzer (Lachat Instruments, Milwaukee, WI, USA). Levels of K, Na, Mg, and Ca were analyzed with an atomic absorption spectrophotometer (Perkin-Elmer 460, Norwalk, CT, USA). Cl was extracted from the leaves in water (100:1 water:dry matter) and the Cl^−^ concentration was determined with a chloride analyzer (Model 926, Sherwood Scientific Ltd., Cambridge, UK).

### 4.8. Statistics

Data in percentages were arcsine-transformed before further analysis. Area under the disease progress curve (AUDPC) values were calculated. Standard errors (SE) of the means were calculated and are presented alongside the degrees of freedom (DF = *n* − 1 for the controlled-conditions experiments and DF = *n* − 2 for correlations calculated for the field data). The disease-severity data and the AUDPC data were analyzed using ANOVA and the Tukey–Kramer HSD test. Statistical analysis was performed (α = 5%) using JMP 14.0 software (SAS Institute, Cary, NC, USA).

To examine the relationship between the concentration of a nutritional element and disease severity in experimental replicates (plots), correlation coefficients were calculated and best-line formulas were calculated using all individual pairs of data (*n*). Correlation line types included linear, exponential, logarithmic, and polynomial. In the captions of the relevant figures, the regression formulas and the Pearson correlation coefficient (*r*) values are presented along with significance levels (*P*) according to degrees of freedom (*n* − 2).

## 5. Conclusions

The severity of cucumber powdery mildew was reduced by the foliar application of Ca, Mg, and K salts, among which SO4^−2^ and Cl^−^ salts both have an effect, but Cl^−^ is somewhat more effective. MKP and polyhalite also provide effective disease control. Increasing the K concentration in irrigated mature plants was effective as well, but the mentioned foliar application seems to provide more effective CPM control. The combination of salts in a single spray solution provided only a minor additive effect and the combination of spray treatments with irrigation treatments did not yield any additive effect. Coupled with the fact that above a certain concentration (measured as the molar concentration of Cl^−^) of the spray-applied salts we saw no additional disease reduction, these findings suggest that induced resistance is a potential mode of action in the CPM pathosystem, as has also been demonstrated in the sweet basil downy mildew pathosystem [18].

## Figures and Tables

**Figure 1 plants-10-02216-f001:**
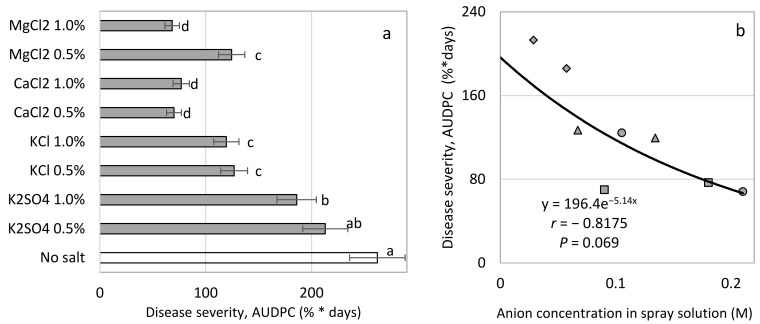
Effects of sprayed applications of salts on the severity of cucumber powdery mildew (CPM, *Podosphaera xanthii*) on leaves (Expt. A-s1). Solutions containing different concentrations of the salts were sprayed on mature plants twice a week. CPM severity was evaluated on a 0 to 100 scale, in which 0 = no disease symptoms and 100 = leaf fully covered by symptoms. (**a**) The area under disease progress curve (AUDPC) was calculated for 18 days from disease onset. Values followed by a common letter are significantly not different from each other according to Tukey-Kramer’s HSD test (*P* ≤ 0.05). (**b**) The relationships between the anion concentrations of the various spray treatments (presented as molar values) and CPM severity were examined. Markers represent MgCl_2_ (●), CaCl_2_ (■), KCl (▲), and KSO_4_ (♦). The regression formula is presented and the Pearson regression (*r*) values are presented along with significance levels (*P*). Bars = SE.

**Figure 2 plants-10-02216-f002:**
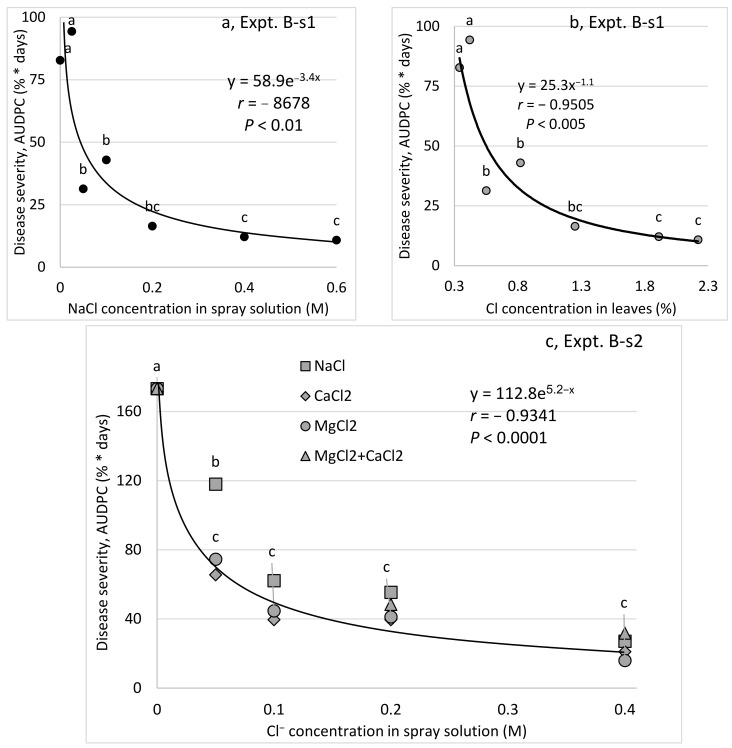
Effects of spray applications of chloride salts of Mg, Ca, and Na on the severity of cucumber powdery mildew (CPM, *Podosphaera xanthii*) on leaves of cucumber plants. Salts were applied to mature plants twice a week. The molar concentrations related to the anion (Cl^−^) are presented. (**a**) Effects of NaCl (applied at concentrations of 0 to 0.6 M in Expt. B-s1) on CPM. (**b**) Effects of the subsequent leaf concentrations of Cl^−1^ on the severity of CPM. (**c**) Effects of MgCl_2_, CaCl_2_, and NaCl (applied at concentrations up to 0.4 M Cl^−^ in Expt. B-s2) on CPM severity. CPM severity was evaluated on a 0 to 100 scale, in which 0 = no disease symptoms and 100 = leaf fully covered by symptoms. The area under the disease progress curve (AUDPC) was calculated for 12 days from disease onset. Values in each concentration followed by a common letter are significantly not different from each other according to Tukey-Kramer’s HSD test (*p* ≤ 0.05).

**Figure 3 plants-10-02216-f003:**
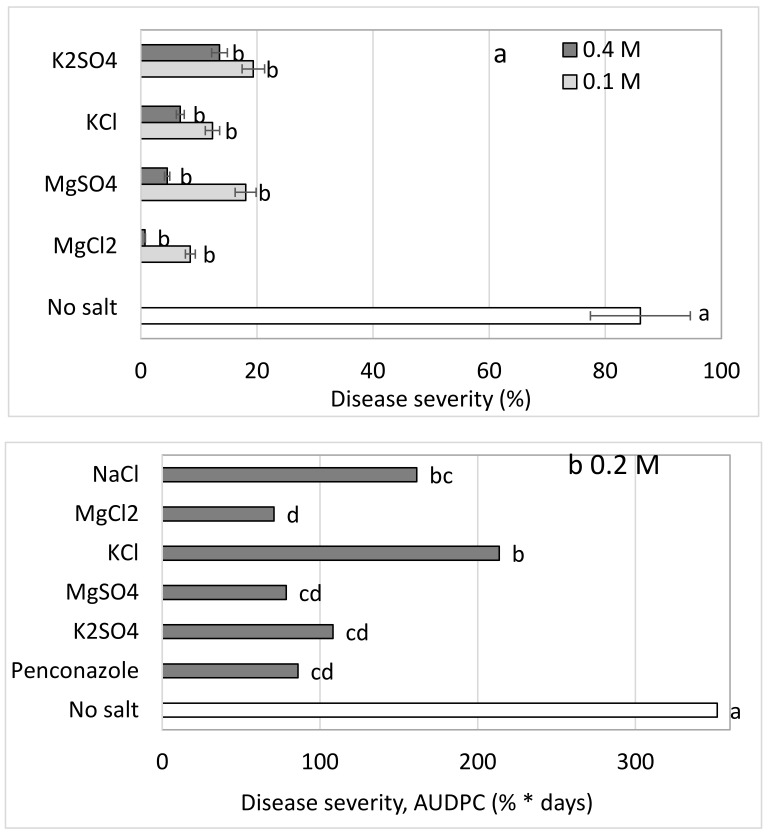
(**a**) Effects on cucumber powdery mildew (CPM, *Podosphaera xanthii*) severity of water (empty, lower column), MgCl_2_, MgSO_4_, KCl, and K_2_SO_4_ sprayed twice a week at concentrations of 0.1 and 0.4 M in Expt. B-s3 and (**b**) a comparison of these salts sprayed at 0.2 M with the chemical fungicide penconazole (0.035% of 200 g/L Ofir, 2000, once a week) in Expt. B-s4. CPM severity was evaluated on a 0 to 100 scale, in which 0 = no disease symptoms and 100 = leaf fully covered by symptoms. The area under the disease progress curve (AUDPC) was calculated. Values for each salt concentration followed by a common letter are significantly not different from each other according to two-way ANOVA with Tukey’s HSD (*p* < 0.0009); the analysis was identical for each set at both concentrations (**a**,**b**). The difference between the two salt concentrations (0.1 vs. 0.4 M, **a**,**b**) was significant (*p <* 0.0009). Bars = SE.

**Figure 4 plants-10-02216-f004:**
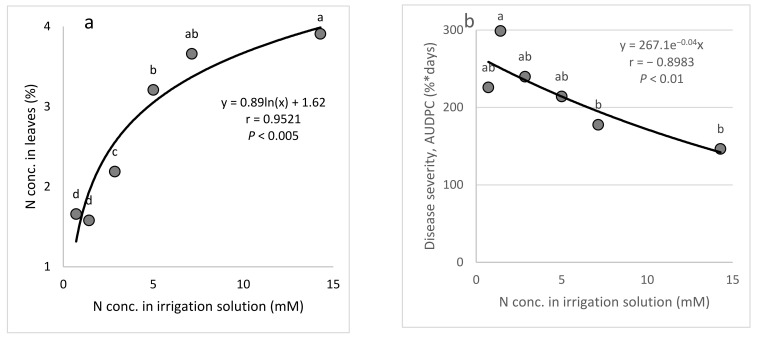
Effect of the N concentration in the irrigation solution on cucumber powdery mildew (CPM, *Podosphaera xanthii,* Expts. B-N-f). (**a**) The N concentration in the leaves relative to the N concentration in the irrigation solution. (**b**) CPM severity relative to the N concentration in the irrigation solution. (**c**) CPM severity relative to the N concentration in the leaves. Disease severity was evaluated on a 0 to 100 scale, in which 0 = healthy leaves and 100 = leaves completely covered by disease symptoms. The area under the disease progress curve (AUDPC) was calculated for 18 days from disease onset. In each graph, values for each N treatment followed by a common letter are significantly not different from each other according to Tukey-Kramer’s HSD test (*p* ≤ 0.05). The regression formulas are presented and the Pearson regression (*r*) values are presented along with significance levels (*p*).

**Figure 5 plants-10-02216-f005:**
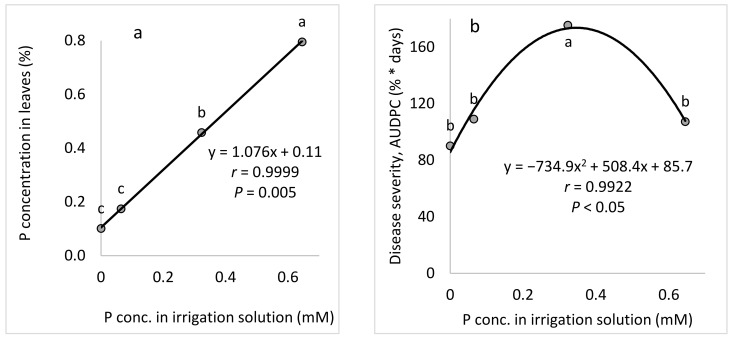
Effects of different concentrations of P in the irrigation solution on cucumber powdery mildew (CPM, *Podosphaera xanthii*, Expts. B-P-f). (**a**) P concentration in the leaves relative to the P concentration in the irrigation solution. (**b**) CPM severity relative to the P concentration in the irrigation solution and (**c**) CPM severity relative to the P concentration in the leaves. Disease severity was evaluated on a 0 to 100 scale, in which 0 = healthy leaves and 100 = leaves completely covered by disease symptoms. The area under disease progress curve (AUDPC) was calculated for 18 days from disease onset. In each graph, values for each P treatment followed by a common letter are significantly not different from each other according to Tukey-Kramer’s HSD test (*p* ≤ 0.05). The regression formulas are presented and the Pearson regression (*r*) values are presented along with significance levels (*p*).

**Figure 6 plants-10-02216-f006:**
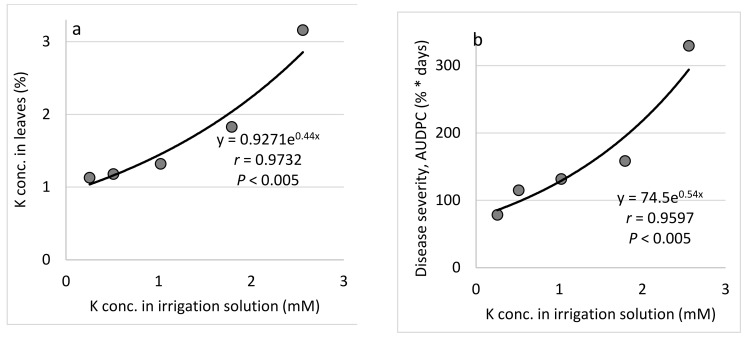
Effect of the concentration of K in the irrigation solution on cucumber powdery mildew (CPM, *Podosphaera xanthii*, Expts. B-K-f). (**a**) K concentration in the leaves relative to the K concentration in the irrigation solution. (**b**) CPM severity relative to the K concentration in the irrigation solution and (**c**) CPM severity relative to the K concentration in the leaves. Disease severity was evaluated using a 0 to 100 scale, in which 0 = healthy leaves and 100 = leaves completely covered by disease symptoms. The area under disease progress curve (AUDPC) was calculated for 18 days from disease onset. In each graph, values for each K treatment followed by a common letter are significantly not different from each other according to Tukey-Kramer’s HSD test (*p* ≤ 0.05).

**Figure 7 plants-10-02216-f007:**
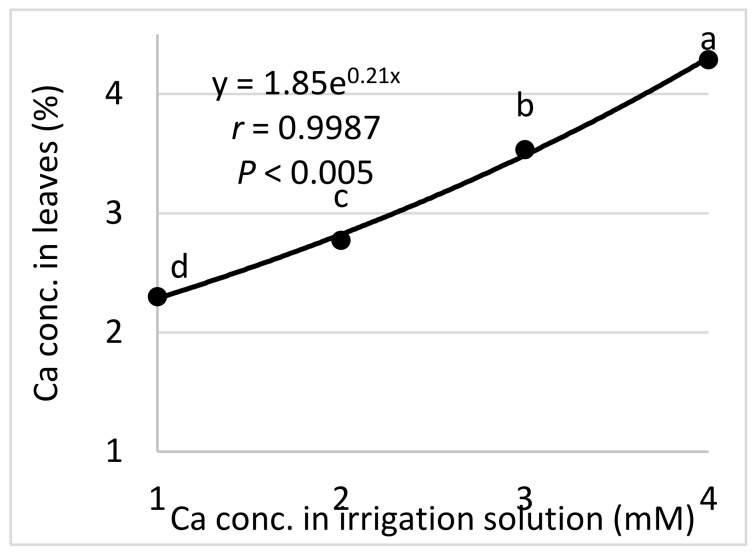
The Ca concentration in the leaves relative to the Ca concentration in the irrigation water (Expts. B-Ca-f). The regression formula is presented and the Pearson regression (*r*) value is presented along with the significance level (*p*); values for each Ca treatment followed by a common letter are significantly not different from each other according to Tukey-Kramer’s HSD test (*p* ≤ 0.05).

**Figure 8 plants-10-02216-f008:**
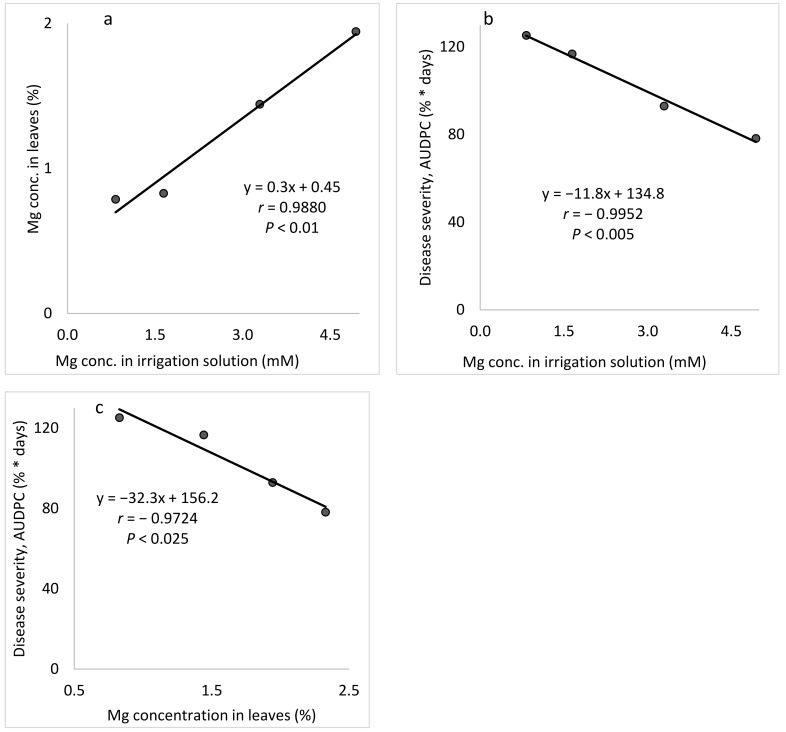
Effect of the concentration of Mg in the irrigation solution on cucumber powdery mildew (CPM, *Podosphaera xanthii*, Expts. B-Mg-f). (**a**) Mg concentration in the leaves relative to the Mg concentration in the irrigation solution. (**b**) CPM severity relative to the Mg concentration in the irrigation solution and (**c**) CPM severity relative to the Mg concentration in the leaves. Disease severity was evaluated on a 0 to 100 scale, in which 0 = healthy leaves and 100 = leaves completely covered by disease symptoms. The area under the disease progress curve (AUDPC) was calculated for 18 days from disease onset. In each graph, values for each Mg treatment followed by a common letter are significantly not different from each other according to Tukey-Kramer’s HSD test (*p* ≤ 0.05). The regression formulas are presented and the Pearson regression (*r*) values are presented along with significance levels (*p*).

**Figure 9 plants-10-02216-f009:**
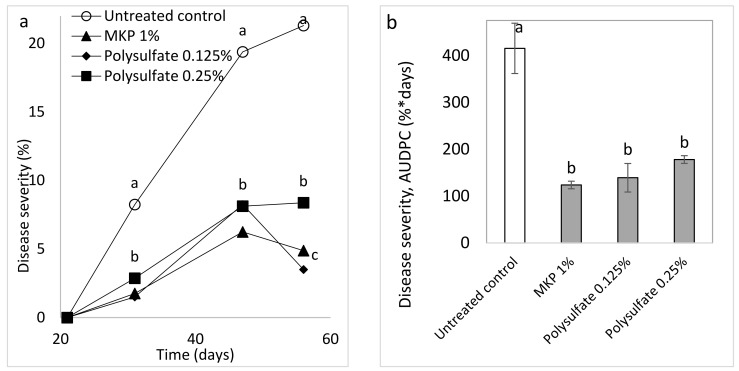
Effect of weekly sprays with polyhalite (polysulfate) and monopotassium phosphate (MKP) on cucumber powdery mildew (CPM, *Podosphaera xanthii*) (Expt. A-SCs-a). (**a**) CPM severity was evaluated on a 0 to 100 scale, in which 0 = no disease symptoms and 100 = leaf fully covered by symptoms. (**b**) The area under disease progress curve (AUDPC) through Day 57. Values for each date and in each graph followed by a common letter are significantly not different from each other according to one-way ANOVA with Tukey’s HSD (*p* ≤ 0.05). Bars = SE.

**Figure 10 plants-10-02216-f010:**
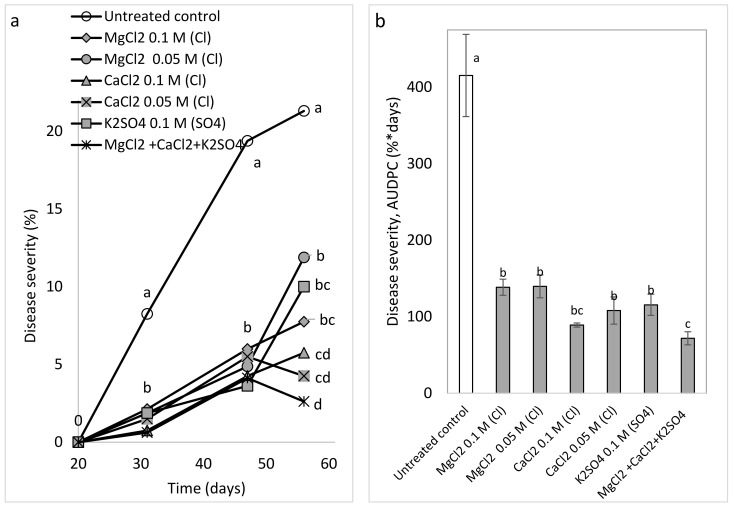
Effect of weekly sprays of various salts at two concentrations, individually and in combination (MgCl_2_ 0.05 M (Cl), CaCl_2_ 0.05 M (Cl), and K_2_SO_4_ 0.1 M (SO_4_)), on cucumber powdery mildew (CPM, *Podosphaera xanthii*) (Expt. A-SCs-b). (**a**) CPM severity was evaluated on a 0 to 100 scale, in which 0 = no disease symptoms and 100 = leaf fully covered by symptoms. (**b**) The area under disease progress curve (AUDPC) was calculated through Day 57. Values for each date and in each graph followed by a common letter are significantly not different from each other according to one-way ANOVA with Tukey’s HSD (*p* ≤ 0.05). Bars = SE.

**Figure 11 plants-10-02216-f011:**
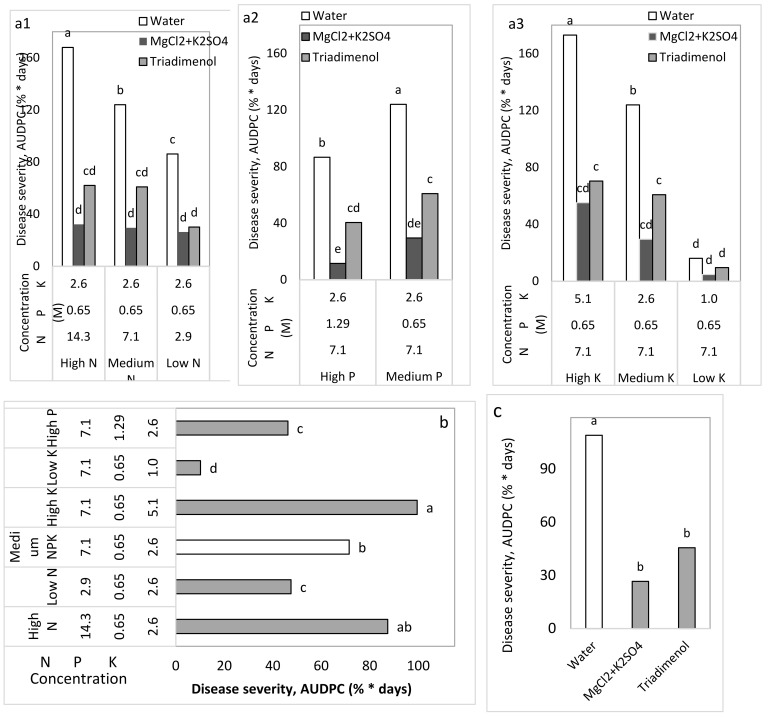
Effects of N, P, and K levels in the irrigation solution and spray applications, every 3 to 5 days, with MgCl_2_+K_2_SO_4_ 0.1 M (Cl^−^) + 0.1 M (SO_4_^−^) and the fungicide triadimenol on cucumber powdery mildew (CPM, *Podosphaera xanthii*) under commercial-like conditions (Expt. CL1). The effects of three N concentrations (**a1**), two P concentrations (**a2**), and three K concentrations (**a3**) in the irrigation solution are presented alongside the three spray regimes (**a1**–**a3**); the medium N, P, and K concentrations are the same in all of the irrigation treatments. CPM severity was evaluated on a 0–100 scale, in which 0 = no disease symptoms and 100 = leaf fully covered by symptoms. The area under the disease progress curve (AUDPC) was calculated through Day 57. Disease severity is described for (**a**) all single combination treatments (irrigation × spray treatments) and for the major treatments either for the (**b**) NPK fertilization levels or (**c**) the spray treatments across irrigation treatments. Values in each graph followed by a common letter are significantly not different from each other according to two-way ANOVA with Tukey’s HSD (*p* ≤ 0.05).

**Figure 12 plants-10-02216-f012:**
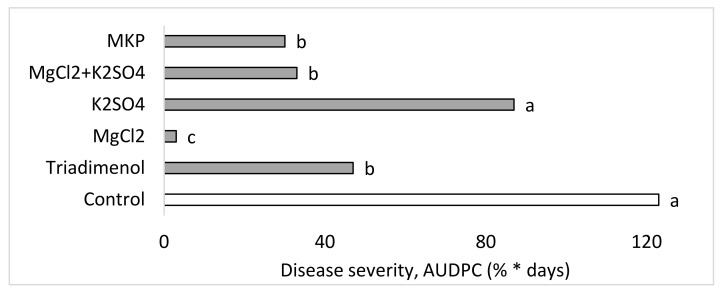
Effects of spray applications of MgCl_2_ 0.1 M (Cl^−^), K_2_SO_4_ 0.1 M (SO_4_^−^),MgCl_2_+K_2_SO_4_, monopotassium phosphate (MKP), and the fungicide triadimenol on cucumber powdery mildew (CPM, *Podosphaera xanthii*) under commercial-like conditions (Ept. CL2). CPM severity was evaluated on a 0 to 100% scale, in which 0 = no disease symptoms and 100 = leaf fully covered by symptoms. The area under the disease progress curve (AUDPC) was calculated through Day 57. Values in each graph followed by a common letter are significantly not different from each other according to one way ANOVA with Tukey’s HSD.

**Table 1 plants-10-02216-t001:** Experimental setup, factors tested, application methods, and growing seasons.

Site	Code	Growing Setting	Materials Tested	Additional Treatment	Application	Season
A	A-s1	Pots	K_2_SO_4_, KCl, MgCl_2_, CaCl_2_		Foliar (spray, “s”)	All year
B	B-s1	Pots	NaCl		Foliar	All year
B	B-s2	Pots	MgCl_2_, CaCl_2_, NaCl, MgCl_2+_CaCl_2_		Foliar	All year
B	B-s3	Pots	MgSO_4_, MgCl_2_, K_2_SO_4_, KCl		Foliar	All year
B	B-s4	Pots	MgSO_4_, MgCl_2_, K_2_SO_4_, KCl	penconazole	Foliar	All year
B	B-N-f	Pots	N		Fertigation (f)	All year
B	B-P-f	Pots	P		Fertigation	All year
B	B-K-f	Pots	K		Fertigation	All year
B	B-Ca-f	Pots	Ca (Cl)		Fertigation	All year
B	B-Mg-f	Pots	Mg (Cl)		Fertigation	All year
A	A-SCs-a	Large pots (semi-commercial)	Monopotassium phosphate (MKP), K_2_Ca_2_Mg(SO_4_)_4_^.^2(H_2_O) (polyhalite as polysulfate)		Foliar	Autumn–Winter
A	A-SCs-b	Large pots (semi-commercial)	MgCl_2_, CaCl_2_, K_2_SO_4_, MgCl_2_+CaCl_2_+K_2_SO_4_		Foliar	Winter–Spring
C	CL1	Boxes (commercial-like)	N, P, K in irrigation (Table 2) X MgCl_2_+K_2_SO_4_	triadimenol	Fertigation and Foliar	Spring
C	CL2	Boxes (commercial-like)	MgCl_2_, K_2_SO_4_, MgCl_2_+K_2_SO_4_, MKP	-”-	Foliar	Spring

**Table 2 plants-10-02216-t002:** Fertigation regimes in the commercial-like experiment (Expt. CL 1).

Treatment		Fertigation Treatments (M)		
1	High N	Medium P 0.65	Medium K 2.6	High N 14.3
2	Low N	Medium P 0.65	Medium K 2.6	Low N 2.9
3	Medium NPK	Medium P 0.65	Medium K 2.6	Medium N 7.1
4	High K	Medium P 0.65	High K 5.1	Medium N 7.1
5	Low K	Medium P 0.65	Low K 1.0	Medium N 7.1
6	High P	High P 1.29	Medium K 2.6	Medium N 7.1

## Data Availability

The data that support the findings of this study are available from the corresponding author upon reasonable request.

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
