# Peer review of "Nutrient Status of Cucumber Plants Affects Powdery Mildew (Podosphaera xanthii)"

_plants, 2021, doi:10.3390/plants10102216_

Round 1

Reviewer 1 Report

This paper presents the results of various experiment on the effects of applications of N, P, K, Mg, and Ca through the irrigation solution and spray applications of K, Ca, and Mg salts on cucumber powdery mildew.

The manuscript need a deep revision as in some part it is not easy to read.

Too many experiments under different and somewhat unreported conditions that brings interesting results but that cannnot offer a clear view on the problem as presented and discussed.

Moreover, even if the results could be of interest the main focus of the manuscript is on the effect on CPM without taking into account the effects of the experimental treatments on plant growth, development and yield. Yield was reported in the results only for commercial like experiments but was not introduced in the material and methods section.

It would be better to refocus the experiments showing also plant response to the treatments. It would be worth to have a limitation of CPM but this cannot help if plant is negatively affected in its physiological and productive aspects.

The experiments should be better linked to show to the reader their consequentiality.

I suggest to deeply revise and refocus the manuscript and add some missing information.

Abstract

The abstract needs to be rewritten to present clearly all the experiments

Introduction

L.55 change microelement with element

Results

L.146-147 This was not clearly reported in the material and methods section.

L.260-261 This information should be added in the material and method section.

L.335-336 This information should be added in the material and method section.

L.335-351 Yield should be reported as fruits/plant or kg/plant

Discussion

In some parts the discussion appears as a review of other works but it doesn't succeed in explicating the major mechanisms underlying the results

L.361-363 erase

L.368-374 This part was already reported in the introduction

L.380 increasing or increased?

L.385 Check and refrase this sentence

L.389-395 It’s not clear the correlation between the protection against F. osysporum mediated by citrate exudation limitation and cucumber powdery mildew

L.378-381; 405-407 The different effects of nutrients on young or mature plants have not been discussed and explained.

Material and Methods

4.1.

This subsection is confusing. Authors should describe the environmental characteristics of the sites and explain when the experiment were performed. The experiments where performed in three sites but this choice is not explained and justified. Environmental conditions could be very different among the sites, affecting plants and pathogen development.

How were the plants spatially arranged in every experiment?

L.532 Is it correct 722g/L?

L.533 Why did you use two different cv. For pot and commercial-like experiments?

L.552 2.2 dS/m? too high!

4.2.

L.557-558 At lines 544-545 authors stated that “For the experiments involving the foliar salt treatments, we used a potting  mixture consisting of coconut fiber:tuff” but here they say that seedlings were transplanted in growth mixture OR perlite. So it’s not clear which was the substrate used in this trial.

L.568-569 No control in experiment A-s1? Why concentrations of the experiment A-s1 are reported as percentage whereas those of the other experiments are reported as M?

4.3.

L.583 Which were the characteristics of perlite and why did you chose this kind of substrate?

L.597 Mo (molybdenum) insted of Mb

L.598 Is it correct NaNO3? Or may be it’s CaNO3? There is no source of Ca among the salts indicated. What about the microelements?

Is not clear how the different concentrations of N P K Ca and Mg were achieved  in order to keep constant all the other elements

Which were EC and pH of the control nutrient solution? The variation of EC and pH in the different nutrient solutions tested  could affect element availability

4.3.1.

L.605  The N concentrations tested should be six as also results from figure 4a;  the concentration 5.7 mM is missing here and is misssing also in figure 4b

4.3.2.

L.613 Check the concentrations of P  as in 4.3. P = 0.35 mM and here is 0.323. Use the same number of digits

L.607-608;615-616;623-624 EC an pH are always the same

4.3.4.

L.630 At line 595 you say 1.3 mM whereas here you report 1.0 mM. Which is correct?

4.3.5.

L.639 At line 595 you say 0.54 mM whereas here you report 0.82 mM. Which is correct?

4.4.

L.645 When, where (open field, greenhouse, environmental conditions)?

L.657 “the upper solubility limit”; please better explain this

4.5.

L.663 Does it was a net house or a greenhouse shadowed with a net? Light, humidity and temperatures inside the greenhouse?

L.681-686 The two concentrations of P are missing

4.6.

This paragraph shold be moved after 4.1

L.703 two different cullitvars are reported in 4.1 whereas here the authors refer to one cultivar

4.7

L.716 When was the harvest time in each experiment?

L.727 add the location of manufacturer

4.8.

The type of the experimental design of each experiment should be reported.

Author Response

Thank you for the review. I followed all comments and corrected the manuscript accordingly. I am sure that thanks to the comments and after the corrections, the manuscript is much improved and it is suitable for publication.

Regards

Yigal

Reviewer 2 Report

Review Report

There are quite a few works in the literature on fungal diseases treated with mineral salts: the results are diverse and sometimes contradictory. The present paper is a comprehensive and well planned research work. It is a applied science work.

Specific comments

In my opinion there is a out of order: the “Materials and Methods” section (line 522) should go as Section 2 instead of Section 4.

I would also suggest a change of place for Table 1 that should go in section 2, perhaps just before table 2. Comment on Table 1, line 554, should change.

Since there are quite a lot of experiments, I believe that a greater clarity in some experimental details and in  some the used terms is needed.

The duration of the experiments is not always mentioned. Sometimes they appear in the legend of figures.

Line 555. Is it any surfactant added in the formulations of any the used sprays?.

Line 660, it says (Expt. CSs b) and 2 lines above (Expt. SCs a).

Lines 643-660. Section 4.4. Are they experiments A or B?. can be

Lines 661-700. Section 4.5. Are all experiments CL are conducted in Site C?.

Line 693, SO4= instead SO4-

Line 697, “four” instead “for”

Line 709-711, It is not clear how the area affected by the fungus, disease severity, is measured.

Results

Line 148. AUDPC (Area under disease progress curve) in Figure 1a, I do not find it is a real curve. Could you find another term?. If treatments went along for 18 days, is the disease progress measured at the end of the experiment?.

What is the meaning of “AUDPC (%*day)”?

Line 150. If CPM severity was evaluated on a 0-100 scale, what is the meaning of a 200% severity?.

Line 154, Figure 1b, what is the reason for markers Mg and Ca, in contrast of KCl and K2SO4?.

Line 175. “Salts were applied ……  “, better in Material and methods.

Line 181. CPM severity is presented for whole plant leaves: is it just in that case?.

Lines 184 y 185, (Results) it is not clear whether they are experiments A or B.

Line 199. Expt. A-s3 should be Expt. B-s3?.

Line 226. Is it 0.46% P instead 0,46 mM (Figure 5c)?

Line 266. Ca effect on CPM severity: Data not shown?.

Line 272. Figure 7. Do the letters a, b, c and d, at the Ca Curve, have any meaning? .

Line 321. SO4= instead SO4-

Line 517.  Improve "......the effects of the anion effect on ......."  

Author Response

(The authors gave the same response as above.)

Round 2

Reviewer 1 Report

The authors of the manuscript  have not provided a point by point response to my comments, and have not fully addressed all the critical points that made me reject the manuscript on the first round of revision. So I cannot modify my first decision. 

Author Response

Thank you for referring again to the manuscript.

The reviewer writes "The authors of the manuscript  have not provided a point by point response to my comments, and have not fully addressed all the critical points that made me reject the manuscript on the first round of revision. So I cannot modify my first decision. "

Sorry, I provided a point to point response and the manuscript was supplied with track changes corrections. In the file that I supply this time, I follow all my answers and corrections. Nevertheless, Three more general corrections made in todays version of the manuscript in order to reflect in the ms text the general limitations of the research. Two of these corrections are:

"Nutritional elements have effects on cucumber plant physiology and development regardless of the pathogen activity while in the current research we focused mainly on the whole outcome of the effect of the nutritional elements on the susceptibility of the plants to CPM". added in the last paragraph of the introduction.

The experiments were conducted under non-controlled conditions so I further clarified "Experiments were conducted under conditions that allowed CPM development Cucurbit powdery mildew." in the M&M

One important comment of the reviewer is that the mechanism is not mentioned. We did not aim at studying the mechanisms but at the end of the discussion and especially in the conclusions we dealt with possible mode of action (induced resistance).

With the many corrections made in the first round and with the new corrections I am sure that we followed all remarks of the reviewer. Note that corrections made the first time were marked YE and the remarks of this round is marked YE2

Sincerely

Yigal Elad 

Round 3

Reviewer 1 Report

There are still some missing or unclear points.

YE: The conditions during the experiments varied. Table 1 gives the timing of the experiments. Variation among sites is not that much significant as Israel is quite small… The locations are given in a better description. The information on experiments design was added.

 YE2: In case the reviewer seeks for actual detailed description of temperatures and relative humidity, it is impossible to supply and it is not important to detail. I placed a clarification un the materials and methods saying that the " Experiments were conducted under conditions that allowed CPM development Cucurbit powdery mildew."

Climatic conditions may strongly vary even between close locations. Temperature an relative humidity are of paramount importance in order to give the opportunity to replicate the experiment to other researchers and to give informations on the conditions in which the treatments were or were not succesfull in controlling CPM.

This was an important issue that has not been addressed

L.533 Why did you use two different cv. For pot and commercial-like experiments?

YE: Bet alpha is the common one that we use in experiments because it lakes resistance traits, including against CPM. The 501 cv in field experiment is one that we could get from a nursery that is known to have CPM under commercial conditions.

These information should be added in the manuscript

L.598 Is it correct NaNO3? Or may be it’s CaNO3? There is no source of Ca among the salts indicated. What about the microelements?

YE: NaNO3 is correct. Ca exists in the water. Tap water originate from desalination plant where Ca is added.

If the water used is from a desalination plant what do you mean saying that Ca was added in the water?  Again, you have to present clear information to the reader so that the experiments could be fully repeatable.

Which were EC and pH of the control nutrient solution? The variation of EC and pH in the different nutrient solutions tested  could affect element availability

YE: The control EC and pH were at the lower end of the scale.

Add this information in the manuscript

Table 2 It’s not clear what the authors mean with “all year”. The pot experiment were performed only once so the season should be defined

L.564 add a reference to audpc calculation method

L.567-568 Still not clear as at lines 537-538 is reported that in the experiments that involved foliar salt treatments a potting mixture consisting of coconut fiber:tuff (unsorted to 8 mm; 7:3 537 vol.:vol.) was used. So, perlite or growth mixture??? It would be better to delete this sentence as the growing media were stated before.